# Multi-Modal Late Fusion Rice Seed Variety Classification Based on an Improved Voting Method

**Xinyi He [1], Qiyang Cai [1], Xiuguo Zou [1], Hua Li [2], Xuebin Feng [2], Wenqing Yin [2] and Yan Qian [1,***

1   College of Artificial Intelligence, Nanjing Agricultural University, Nanjing 210031, China
2   College of Engineering, Nanjing Agriculture University, Nanjing 210031, China
*   Correspondence: qianyan@njau.edu.cn; Tel.: +86-25-5860-6585

**Abstract:** Rice seed variety purity, an important index for measuring rice seed quality, has a great impact on the germination rate, yield, and quality of the final agricultural products. To classify rice varieties more efficiently and accurately, this study proposes a multimodal l fusion detection method based on an improved voting method. The experiment collected eight common rice seed types. Raytrix light field cameras were used to collect 2D images and 3D point cloud datasets, with a total of 3194 samples. The training and test sets were divided according to an 8:2 ratio. The experiment improved the traditional voting method. First, multiple models were used to predict the rice seed varieties. Then, the predicted probabilities were used as the late fusion input data. Next, a comprehensive score vector was calculated based on the performance of different models. In late fusion, the predicted probabilities from 2D and 3D were jointly weighted to obtain the final predicted probability. Finally, the predicted value with the highest probability was selected as the final value. In the experimental results, after late fusion of the predicted probabilities, the average accuracy rate reached 97.4%. Compared with the single support vector machine (SVM), k-nearest neighbors (kNN), convolutional neural network (CNN), MobileNet, and PointNet, the accuracy rates increased by 4.9%, 8.3%, 18.1%, 8.3%, and 9%, respectively. Among the eight varieties, the recognition accuracy of two rice varieties, Hannuo35 and Yuanhan35, by applying the voting method improved most significantly, from 73.9% and 77.7% in two dimensions to 92.4% and 96.3%, respectively. Thus, the improved voting method can combine the advantages of different data modalities and significantly improve the final prediction results.

**Keywords:** rice seed; variety classification; multimodal fusion; machine vision; point cloud

## 1. Introduction

As the most primitive and fundamental means of production in agricultural development, seeds not only determine the survival rate and growth activity of seedlings but also affect subsequent product processing. In agricultural production, with improvements in the production capacity and product quality requirements of various crops, effectively selecting and breeding good varieties has become a hot research topic.

Machine vision research is the process of processing visual information, usually including the image brightness, shape, position, color, and texture. Using machine vision to classify varieties can achieve the effect of nondestructive testing, so it has become a good research direction in recent years. Initially, the variety detection of rice seeds started from 2D images. In [1], an automatic rice quality evaluation system based on an artificial neural network (ANN) and support vector machine (SVM) classifiers was proposed. Experiments showed that the overall accuracy of the proposed ANN classifier was 83%, while that of the SVM was 91%. In [2], rice varieties were classified according to color, shape, and texture characteristics. Principal component analysis (PCA) was used to reduce the dimension of the data. Using discriminant analysis (DA), the accuracy of segregation of rice, brown rice, and white rice cultivars was 89.2%, 87.7%, and 83.1%,

respectively. To identify and classify the desired species, a multilayer perceptron neural network was implemented based on the most effective components. The results showed that the network was 100% accurate in identifying and classifying all of the mentioned rice varieties. In [3], using seven morphological features extracted from each variety of rice, a model was created by using LR (logistic regression), MLP (multilayer perceptron), SVM, DT (decision tree), RF (random forest), NB (naïve Bayes), and weighted k-nearest neighbor (kNN) machine learning techniques, and the performance measurement values were obtained. The experimental results showed that the classification accuracy rates of the models were 93.02% (LR), 92.86% (MLP), 92.83% (SVM), 92.49% (DT), 92.39% (RF), 91.71% (NB), and 88.58% (kNN). As a branch of machine learning, neural networks are gradually being widely used. A new method using a deep convolution neural network (CNN) as a general feature extractor was proposed in [4]. The extracted features were classified using an ANN, cubic SVM, quadratic SVM, kNN, boosted tree, bagged tree, and linear discriminant analysis (LDA). Compared with a model based on simple features, the model trained with CNN-extracted features showed better classification accuracy. The CNN-ANN classifier showed the best performance. The classification accuracy was 98.1%, recall 98.1%, and F1-score 98.1%, in 26.8 s. In [5], the authors proposed a seed classification system based on CNN and transfer learning, which contained models and used advanced deep learning techniques to classify 14 common seeds. The techniques used in that study included the decayed learning rate, model checkpointing, and hybrid weight adjustment. The proposed model exhibited 99% recognition accuracy for 234 training and testing images.

Compared with simple two-dimensional (2D) image recognition, the three-dimensional (3D) information obtained from the surface of rice seeds can describe the seed appearance more completely and accurately. However, the application of 3D computer vision in rice seed modeling is still at the research stage, and its implementation in crop seed modeling and nondestructive testing (NDT) is still being popularized on a small scale. In [6], a rice variety classification method based on 3D point cloud data of the rice seed surface and a deep learning network was proposed. The preprocessed point cloud was input into the improved PointNet network for feature extraction and variety classification. The average classification accuracy of the improved PointNet model for eight rice varieties was 89.4%. In [7], a rice seed recognition platform was constructed by combining 3D laser scanning technology and the BP neural network algorithm. Information on the rice seed surface was collected from four angles, and three morphological characteristics and projection characteristics of the main plane cross-section were obtained by feature calculation. The results showed that for input vectors composed of nine surface morphological features in 3D, the recognition rates of five rice varieties were 95%, 96%, 87%, 93%, and 89%, respectively. The recognition rates for the input vectors composed of nine projective features of the rice seed cross section were 96%, 96%, 90%, 92%, and 89%, respectively. The 3D grain character measurement method based on CT was studied in [8]. Here, 3D rice spike images were reconstructed by 3D reconstruction software, and grain phenotypes were analyzed. The results show that the recognition accuracy of a random forest classifier was higher than that of an LDA classifier and SVM classifier, and the average cognition accuracy was 95.19%.

The 2D and 3D models provide complementary information. Each pixel of an RGB image obtains various colors by changing the three color channels of red (R), green (G), and blue (B) and superimposing them. In 2D, the original image collected by the camera is an RGB image. RGB images have a higher resolution than the depth images or point clouds and contain rich textures not available in the point domain. In addition, images can cover "blind spots" caused by reflective surfaces that depth sensors cannot perceive. In contrast, 2D images are limited in 3D detection tasks because they lack absolute object depth and scale measures, which can be provided by 3D point clouds.

Multimodal technology helps artificial intelligence understand the external world more accurately by cooperating with perceptual information in multiple modalities. According to the chronological order of fusion, the methods for merging 2D images and 3D

point clouds can be divided into two types: early fusion and late fusion. Early fusion fuses the features extracted from different modalities, which is also called feature fusion [9]. In [10], the authors proposed a fusion method combining RGB and depth information. The model consisted of a two-stream CNN that can automatically fuse information from RGB and depth using a specific encoding method before classification. Finally, the goal of learning rich features from two domains was achieved. The authors of [11] proposed a method for fruit leaf disease classification based on feature fusion. They used transfer learning to adjust the extracted deep features and then fused multiple features into the final feature through a multilevel fusion algorithm based on entropy-controlled threshold calculation. The fused features were input into a main classifier multi-SVM. The experimental results showed that the method improved the recognition accuracy (97.8%) and sensitivity (97.6%) of the five diseases. The authors in [12] used partial least squares (PLS) regression to perform feature selection from the extracted deep feature set. The acquired features were input into the ensemble baggage tree classifier to realize the automatic disease identification of tomato, potato, and corn crops. The accuracy rate was approximately 90.1%. In [13], they proposed a corn seed variety detection method that weighted the data at different stages after the feature extraction of corn seed images and then fused the shallow features with the deep features to construct multiscale fusion features. Experiments showed that the average precision of the MFSwin Transformer model on the test set was 96.53%, which was higher than that of the other models. Late fusion fuses the prediction scores of multiple single modalities, also known as the score fusion [9]. The authors in [14] proposed a weed classification method by multimodal late-fusion deep neural networks (DNNs) using a Bayesian conditional probability-based method, or determining the priority weights to calculate the score vector. The results showed that the method was effective in plants. The accuracy rate on a seedling dataset was 97.31%. The study in [15] proposed a method to estimate the ripeness of papaya fruit by combining hyperspectral and visible light images, enabling multimodality through the late fusion of image-specific networks. Experimental results showed that the model obtained an improved F1-score of up to 0.97. The compatibility of early fusion and late fusion is relatively good, and this approach can adapt to most detection algorithms based on point clouds. However, when there is a problem with the classification model in early fusion, the correct detection of the variety can no longer be achieved. For late fusion, the result is fused by the classification results of multiple models, so this approach is less affected by a single model and is more robust.

It can be seen from the previous research results that most of the research on variety detection in the past only stayed in a single 2D or 3D, and did not combine the advantages of the two modes. In addition, there is a lack of effective weights to predict the outcome during late fusion. Therefore, on the basis of our predecessors, we proposed a new experimental approach. In this study, 2D RGB images and 3D point cloud data captured by Raytrix light field cameras were used as input data for recognition, and an improved voting method was used to fuse the recognition results. We pursued this research goal as follows. (1) The data of the 2D rice pictures and 3D point cloud sets were established and divided into a training set and a validation set at a ratio of 8:2. (2) SVM, kNN, CNN, and MobileNet were used to classify the 2D images. PointNet models were used to classify the 3D point clouds. Finally, a voting method was used to fuse the classification results of multiple models to obtain the final variety detection results. (3) The final classification results were evaluated and compared with the general model classification results through visualization. For the results of this study, we propose the following hypotheses. (1) Multimodal fusion to achieve variety detection can achieve higher accuracy by combining the advantages of 2D and 3D. (2) The late-fusion method can improve the accuracy and robustness of the classification process. Moreover, when the homogeneity of multiple classification models is smaller, the accuracy of the final fusion result is higher.

## 2. Materials and Methods

### 2.1. Point Cloud Collection System

The rice seed 3D point cloud acquisition hardware platform included a camera and lens, a vertical lifting device, a light source and its control module, and a camera calibration tool. There were four parts in total. The research-grade light field camera was model R42, manufactured by the German Raytrix, with a maximum resolution of 41.5 MegaRays and 7708 × 5352 pixels. The imaging lens used in the acquisition process was a 3D light field lens with a focal length of 50 mm and an aperture of F/2.80. The camera and lens were composed of a light field camera and an imaging lens. The computer was equipped with a high-performance GPU, model NVIDIA GTX 1080, for real-time light field processing. The camera calibration tool is a calibration board evenly covered with a dot matrix with a pitch of 2.09 mm. The point cloud acquisition hardware platform collects 3D point cloud data on the surface of rice seeds through cameras and lenses and uses vertical lifting devices to realize rough adjustment and fine adjustment of the height of the camera placement. Camera calibration tools are used to perform light field camera calibration, and the light source control module controls the environment of the experimental site. The experimental environment for point cloud collection is shown in Figure 1.

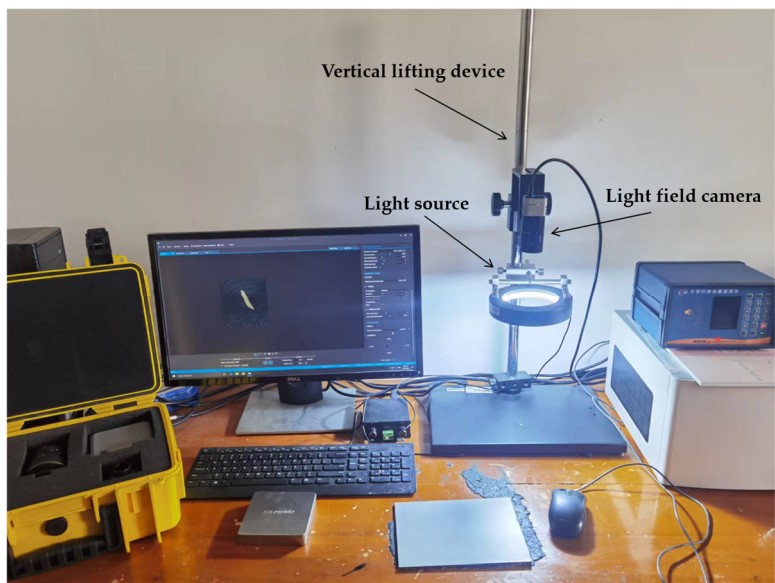

**Figure 1.** Point cloud collection hardware platform.

The camera operation software supports RxLive4.0 software. The software can identify the connected camera, control it, and change the camera parameters. The light field camera can be calibrated using the camera calibration module in the software, with the function of evaluating the calibration effect and evaluating the grade of the calibration result. The software includes a point cloud preprocessing function, which can perform filtering, noise reduction, sharpening, smoothing, and cropping on the point cloud. The data export function can set various file types, file naming formats, and export file storage locations for data export.

### 2.2. Dataset Preparation

In this study, eight common rice seeds in China were selected as the datasets for model training and testing. The seeds selected for the experiment were preliminarily screened and cleaned manually to avoid irregularities such as attached impurities and gaps from affecting the classification results. The storage of the selected samples strictly follows the standardized storage environment. Rice seeds were stored in a dry, low temperature, and airtight environment to avoid being affected by the external environment. The seeds

included in the dataset were Nanjing9108, Zhenghan10, Hannuo35, Yuanhan35, Liannuo13, Hyou518, Huanghuazhan, and Liusha, and the representative RGB images of each category are shown in Figure 2.

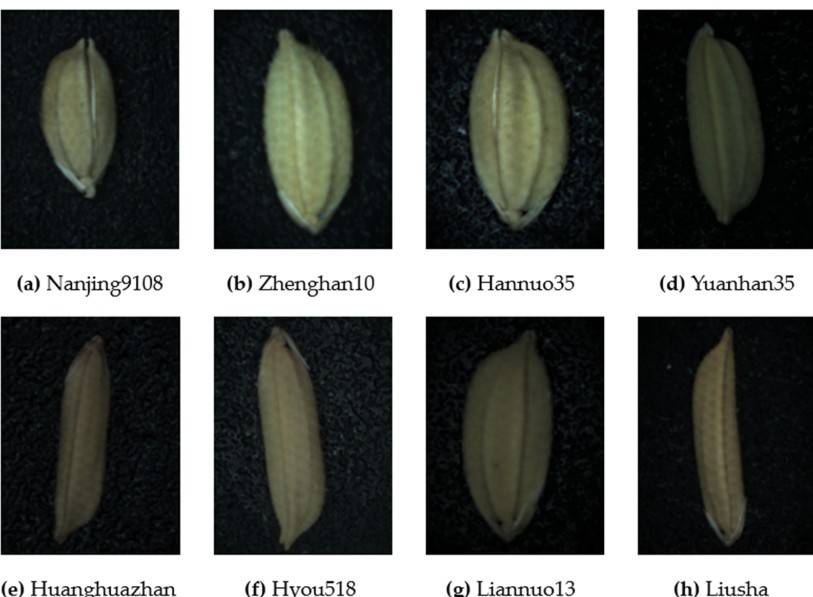

**(a)** Nanjing9108      **(b)** Zhenghan10      **(c)** Hannuo35      **(d)** Yuanhan35

**(e)** Huanghuazhan      **(f)** Hyou518      **(g)** Liannuo13      **(h)** Liusha

**Figure 2.** RGB images of the maize seed grains. (**a**) Nanjing9108. (**b**) Zhenghan10. (**c**) Hannuo35. (**d**) Yuanhan35. (**e**) Huanghuazhan. (**f**) Hyou518. (**g**) Liannuo13. (**h**) Liusha.

The above-mentioned seeds were placed separately on the camera stage to sequentially collect the 2D images and 3D point clouds of the front and back sides. After simple preprocessing of the 2D image, the redundant background was cut according to the size and shape of the seed. Establishing the 3D point cloud first requires preprocessing operations such as denoising, smoothing, and cutting on the original data. However, the rice seed point cloud collected at this time still contains considerable redundant data. Storing, processing, and displaying these point cloud data would increase the burden on the computer processing process and at the same time occupy a large amount of computer hardware and software resources, reducing the efficiency of the operation process. However, if the point cloud is too small, it will lose its features for classification. Therefore, this study downsampled the point cloud to a scale of 2048 points. The final point cloud data after processing are shown in Figure 3.

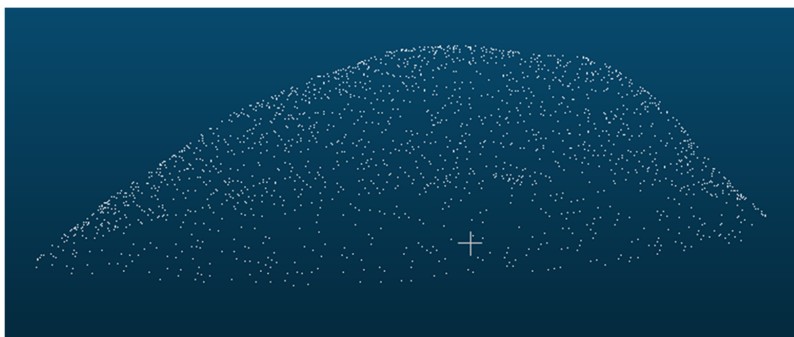

**Figure 3.** The point cloud data after processing.

In the experiment, the dataset of each rice species was divided into a training set and a test set at a ratio of 8:2. Each seed had a corresponding 2D picture and 3D point cloud on the front and back. There were eight varieties for a total of 3194 samples; the total size of the training set was 2560, and the total size of the test set was 634, as shown in Table 1.

**Table 1.** Rice seed dataset.

| No. | Cultivar Name | Training Set | Validation Set |
|---|---|---|---|
| 1 | Nanjing9108 | 320 | 79 |
| 2 | Zhenghan10 | 320 | 79 |
| 3 | Hannuo35 | 320 | 79 |
| 4 | Yuanhan35 | 320 | 80 |
| 5 | Huanghuazhan | 320 | 80 |
| 6 | Hyou518 | 320 | 78 |
| 7 | Liannuo13 | 320 | 80 |
| 8 | Liusha | 320 | 79 |
| | Total | 2560 | 634 |

*2.3. Classification Model*

For the voting method, we need to consider the possible impact of different base models. In theory, the base model can be any model that has been trained. However, in practical applications, if the voting method is to produce better results, two conditions need to be met:

1. The effect between the base models cannot be too different. When a base model performs poorly relative to other base models, the model is likely to be noisy.

2. There should be less homogeneity among the base models. For example, when the prediction effect of the base model is similar, voting based on a tree model and a linear model is often better than voting based on two tree models or two linear models.

Based on the above principles, this paper selected SVM, kNN, CNN, and MobileNet as the base models for 2D classification and point net as the base model for 3D detection.

The SVM is a method of machine learning that was developed based on statistical theory [16]. When SVM is used for classification, it can achieve better classification results when the number of training samples is smaller. Different varieties of rice seeds have large differences in color from the appearance point of view, so an image histogram can be used for classification. First, the image is scaled to a uniform size, and then the image histogram is calculated, as shown in Figure 4. The histogram can be used to obtain the number of pixels of each brightness level of the image of each sample, and displays the distribution of the pixels in the image. The SVM was used to process the image histogram, and the kernel function was linear. The trained model was used to predict the test set, and the prediction probability corresponding to each seed was saved as the input data of the final voting method.

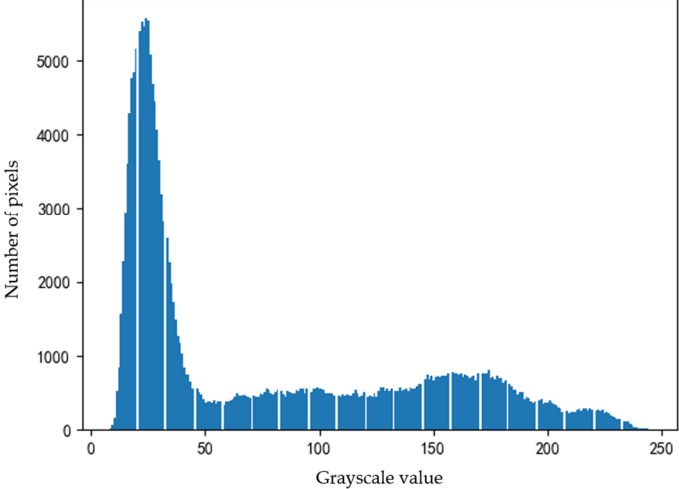

**Figure 4.** Calculated image histogram.

In machine learning, the kNN algorithm is a widely used classification and regression method [17]. This algorithm determines the similarity of the samples to be tested according to the distance characteristics of the nearest neighbor samples to classify them; that is, the category of the samples to be tested is determined by calculating the distance between the sample to be tested and the k-nearest neighbor samples in the training set. The three basic elements of the kNN algorithm are the distance measure, the selection of the k value of the number of neighbors, and the classification decision rule. The histogram calculated according to the training set is input into the kNN model with the neighbor value k parameter of 11 for training to obtain the predicted probability.

The CNN, a type of neural network, is one of the best algorithms for image content, and it performs very well in related operations such as image segmentation, classification, detection, and retrieval. CNNs are structurally composed of multilayer networks, and each layer can be regarded as a plane composed of independent neurons. The main layers include the input layer, convolutional layer, pooling layer, fully connected layer, and classifier [18]. In this experiment, the CNN model was first normalized, and numbers between 0–255 were normalized to between 0 and 1. Then, the output was set to a convolutional layer with 32 channels; the size of the convolution kernel was 3 × 3, and the activation function was ReLU. Then, a pooling layer was added, with a pooled kernel size of 2 × 2. Then, the output was set to a convolutional layer with 64 channels; the convolution kernel size was 3 × 3, and the activation function was ReLU. Another pooling layer was added to perform a pooling operation on a 2 × 2 area. Finally, the 2D output was converted into one-dimensional output through the softmax function to output the model to the neuron of the class name length, and the activation function adopted the corresponding probability value of softmax. The specific network architecture of the CNN is shown in Figure 5.

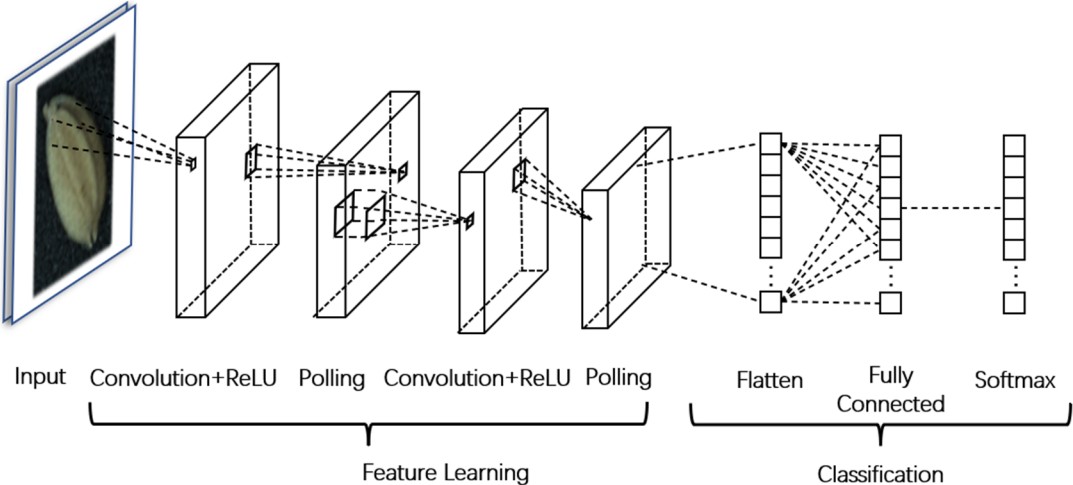

**Figure 5.** CNN network structure.

Compared with the CNN, MobileNet abandons the traditional convolution and combines depth-wise convolution and pointwise convolution as the basic network module [19]. We call this approach depth-wise separable convolution. It uses a very simple stacking structure that has the advantage of improving network computing efficiency and reducing the number of parameters. In this study, we first loaded the pretrained MobileNet model as the backbone model and normalized the input image. Then, the output of the backbone model was the global average pooled and mapped to the final classification number through the fully connected layer.

PointNet is a pioneering approach to feeding point cloud data directly into neural networks. The framework mainly solves the problems of point cloud disorder and permutation invariance. Considering the disorder of the point cloud, PointNet does not convert the point cloud into a multi-view or voxel grid but processes the points directly. For permuta-

tion invariance, this method used a multilayer perceptron to extract features independently for each point and then uses the maximum pooling layer to aggregate the information of all points to obtain global features. In addition, the framework adds T-Net to spatially align the input point cloud and its features by constructing a transformation matrix to solve the problem of transformation invariance. This study used the basic PointNet network model as the classification model, and its structure is shown in Figure 6. After inputting the point cloud data, T-Net was first performed for affine transformation, which was specifically expressed as multiplying the transformation matrix by $3 \times 3$, and then feature extraction was performed through the convolutional layer. According to the model structure, the number of convolution kernels of the two MLP convolutional layers (64, 64) was 64. The convolution kernel size of the first layer of convolution was $1 \times 3$, and the second layer was a $1 \times 1$ kernel. Then, the same feature transform was performed, and in the next MLP (64, 128, 1024), the size of the convolution kernel was $1 \times 1$. After the pooling layer, three fully connected layers were connected, and the number of output nodes was 512, 256, and k in turn. Finally, the softmax function was used to obtain the result.

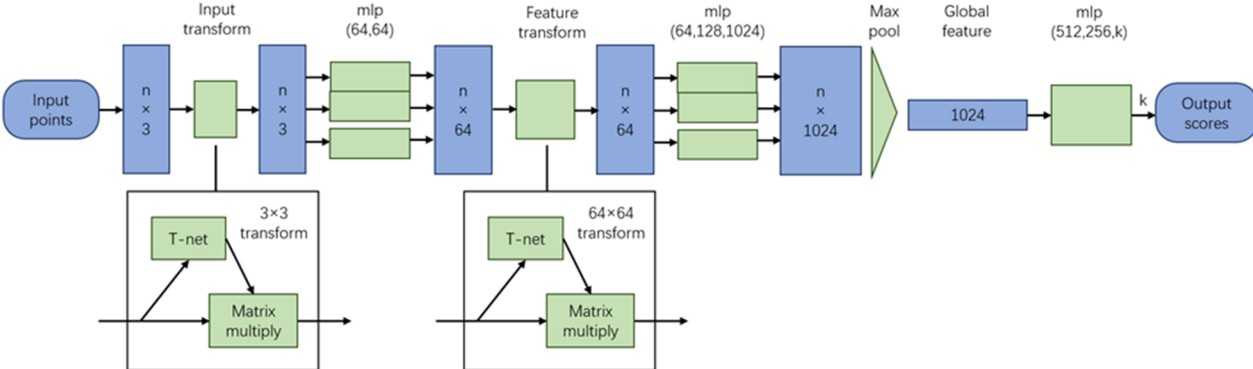

**Figure 6.** Basic PointNet model structure used in this article.

After the above model training was completed, the results were used to predict the data of the test set and output the predicted probability of each sample separately as the input data of the voting method for late fusion.

### 2.4. Improved Voting Method

The voting method is a commonly used technique in ensemble learning that can improve the generalization ability of the model and reduce the associated error rate. The traditional voting method follows the principle of the minority obeying the majority and integrates multiple models to reduce the variance and improve model robustness. Ideally, the forecasting performance of the voting method should be better than that of any one of the base models. When the voting method is applied to the classification model, its prediction result is the most frequent prediction result among all models. According to different prediction methods, classification voting can be divided into hard voting and soft voting. Hard voting simply counts the most common class among all model predictions as the final result. Soft voting calculates the sum of the probability values of the prediction results of each model and selects the class with the highest probability value as the final result. Compared with the hard voting method, the soft voting method takes into account the additional information of the prediction probability, enabling it to obtain more accurate prediction results than the hard voting method.

The traditional voting method has certain limitations: it treats all models the same. That is, for the voting method, all models contribute equally to the prediction. Vote predictions can be biased if some models are good in certain situations but poor in others. Therefore, this study presents an improved soft voting method, which improves the process of calculating the arithmetic mean in the traditional method. This method determines the weights of different models according to their performance scores, combines the probabili-

ties of the predicted result classes of each model to obtain the comprehensive score vector of each model, and uses this vector to determine the predicted results. The late-fusion process based on the improved voting method is shown in Figure 7.

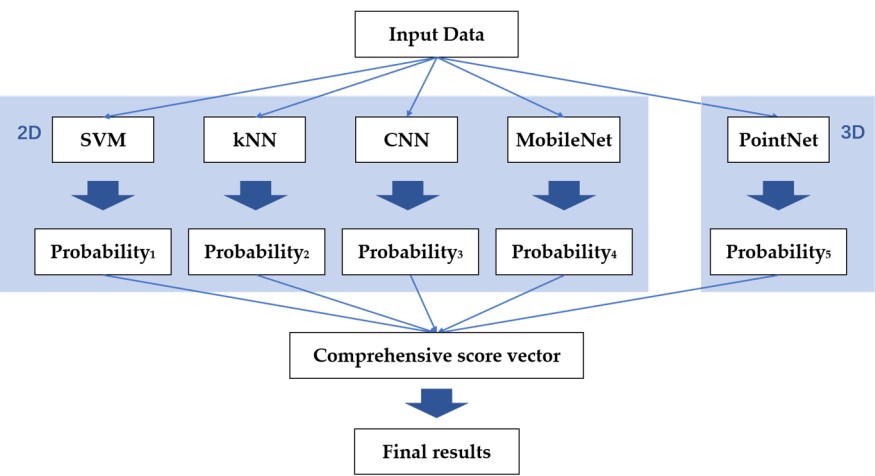

**Figure 7.** Flowchart of the improved voting method.

The F1-score is an indicator used to measure the accuracy of a binary classification model. It takes into account both the precision and recall of the classification model. Later, the traditional F1-score was extended to a multicategory F1-score, which can be divided into macro-F1 and micro-f1 according to the suitable dataset. Macro-F1 is applicable to the classification situation where each category has equal status and the same size [20]. Since the size of the dataset of each variety in this experiment is the same, the Macro-F1 value can be used to determine the scoring vector of each model.

In the binary classification problem, it is assumed that the sample has two categories: positive and negative. When the classifier prediction ends, we can divide the classification results into the following categories:

- True positive (TP): Positive samples are successfully predicted as positive.
- True negative (TN): Negative samples are successfully predicted as negative.
- False positive (FP): Negative samples are incorrectly predicted as positive.
- False negative (FN): Positive samples are incorrectly predicted as negative.

In the binary classification problem, the calculation method of the F1-score is as follows:

$$\text{Precision} = \frac{TP}{TP + FP} \tag{1}$$

$$\text{Re}call = \frac{TP}{TP + FN} \tag{2}$$

$$F1 - score = \frac{2 \times \text{Precision} \times \text{Recall}}{\text{Precision} + \text{Recall}} \tag{3}$$

The F1-score can balance the two indicators of precision and recall at the same time, so it can be used to reflect the classification performance of the model. To extend the calculation method in the binary classification problem to the multiclassification problem, each category can be regarded as a binary classification problem, and the precision and recall can be calculated separately. Therefore, for each class, the distribution of its results in the confusion matrix is as shown in Figure 8.

|  | Predicted | | |
|---|---|---|---|
| | Class1 | Class2 | Class3 |
| Class1 | | | FP |
| Class2 | | | FP |
| Class3 | FN | FN | TP |

**Figure 8.** Distribution of TP, FP, and FN in multiclassification problems.

Formulas (1) and (2) were used to calculate the respective precision, recall, and F1-score in each category, respectively, denoted as $P_1, P_2, \ldots, P_n$; $R_1, R_2, \ldots, R_n$; and $F_{11}$, $F_{12}, \ldots, F_{1n}$. Since the datasets of each variety have the same size, the overall Macro-F1 calculation method is as follows:

$$Macro - F1 = \frac{\sum_{i=1}^{n} F1_i}{n} \qquad (4)$$

Macro-F1 calculated by each multiclassification model is used as the weight of voting to form a scoring vector, which is recorded as $S_1, S_2, \ldots, S_n$. Assuming that there are m varieties in total, the probability of each model predicting that the result is m is $M_1, M_2, \ldots, M_n$. Then, the final probability (Final-P) of each variety predicted by the voting method is:

$$Final - P = \frac{\sum_{i=1}^{n} M_i \times S_i}{n} \qquad (5)$$

The variety corresponding to the highest probability is selected as the final prediction result.

## 3. Results and Discussion

First, we used 2D pictures as input datasets to train SVM, kNN, CNN, and MobileNet.

For both the CNN and MobileNet, 30 epochs are allowed to pass. The accuracy and loss changes of the training set and validation set in each epoch of the two models are represented by line graphs, as shown in Figure 9.

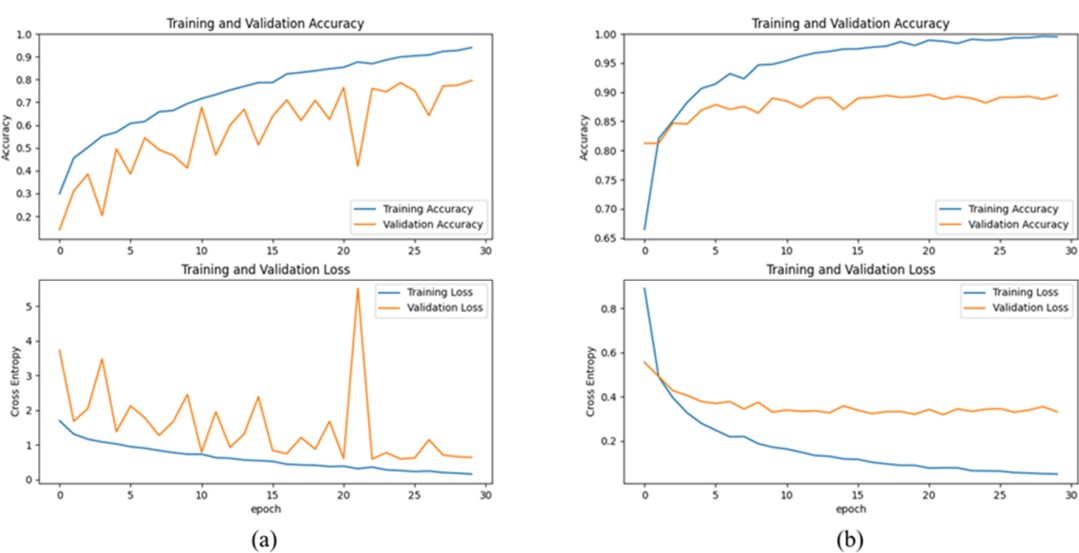

**Figure 9.** Accuracy and loss of the training set and validation set for each epoch: (**a**) CNN, (**b**) MobileNet.

In the early stage of training (the first ten epochs), the overall accuracy of the model gradually increases, and the loss gradually decreases. However, the two indicators of the CNN's validation set fluctuate greatly, while those of MobileNet change steadily with small fluctuations. At the twentieth epoch, the accuracy and loss of the CNN continue to fluctuate greatly. It may be that the appearance similarity of some rice varieties is relatively high, which affects the classification effect of the model but then gradually reduces the fluctuation range. Both curves tend to be smooth. Compared with the CNN, the change in MobileNet is more stable. The convergence is basically completed at the 15th epoch, and the curve basically fluctuates over a small range only. A comparison of the two neural networks revealed that the training process of CNN fluctuated greatly, while MobileNet was relatively stable. After the final stabilization, the accuracy of the CNN reached 79%, and the loss was 0.64. The accuracy of MobileNet was 89%, and the loss was 0.33. Although the accuracy of the CNN was relatively low, this did not have a negative impact on the final result because the prediction probabilities of multiple models need to be fused during late fusion.

We input the 3D coordinates of the processed point cloud data into the PointNet model; Figure 10 shows the accuracy and loss variation during the steps of the training process. From 0 to 2000 steps, the accuracy and loss varied greatly, and the curves were very steep. After 6000 steps, both the accuracy and loss gradually converged, basically fluctuating over a small range only. This change showed that the result of this training was convergent, and the final average accuracy of PointNet was 88.75%. Compared with other models, the recognition accuracy of PointNet was not very high. The reason may be that the method of downsampling during dataset preprocessing is not effective enough, with many important feature points screened out in the process. Therefore, the final classification effect was affected to a certain extent.

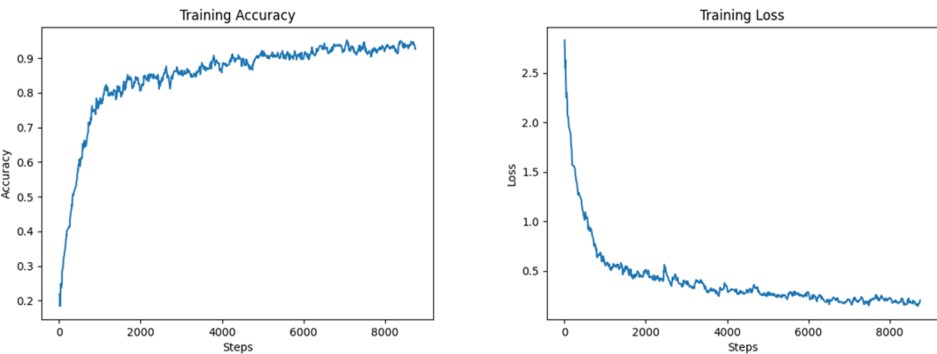

**Figure 10.** Changes in the accuracy and loss of the training set with the step size.

After all of the model training had been completed, we uniformly used the test set for testing and calculated the accuracy of the different models for different types of seeds, as shown in Figure 11. The four models for classifying 2D images had high recognition accuracy for Huanghuazhan and Liusha, and the recognition accuracy reached 98% and 95%, respectively. Moreover, the recognition accuracy of the four 2D models for Huanghuazhan was maintained over the extremely small range of 96% to 99%, indicating that each model has a good classification effect on this variety. However, for Hannuo35 and Yuanhan35 rice seeds, the recognition accuracy for the 2D images was relatively low, and the accuracy of the CNN for Hannuo35 was the lowest, only 52%. The reason may be the appearance similarity of these two kinds of seeds; the currently proposed model may not be adaptable enough to them. In the 3D point cloud recognition, the classification accuracy for each species of seeds using the PointNet model indicates that the difference in the recognition accuracy for the eight seeds was small, with the values remaining between 85% and 95%. The recognition accuracy for Zhenghan10 was relatively low, at 85%. Liusha had the highest recognition accuracy, reaching 92%. Compared with the accuracy for some varieties in 2D, which was notably low, the recognition effect for 3D was higher, and the

values were all maintained at a high level. This shows that the point cloud data can better distinguish the eight kinds of seeds, especially Hannuo35 and Yuanhan35, which cannot be classified correctly in 2D. Therefore, the 3D point cloud data can be used as an effective supplement to 2D classification results.

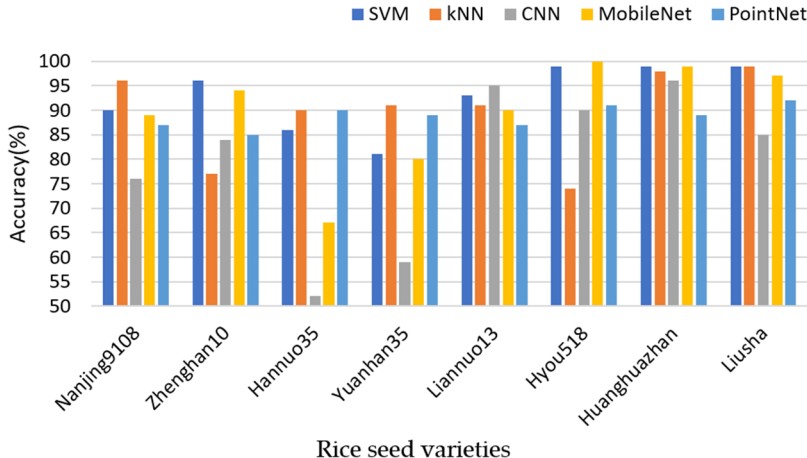

**Figure 11.** Accuracy of the different models for each species of rice seeds.

According to the prediction results, we can build a confusion matrix for each algorithm separately, as shown in Figure 12. TP, FP, and FN of the model can be calculated through this matrix, and the corresponding precision, recall, and Macro-F1 can be calculated based on these parameters. Macro-F1 is used as the weight for late fusion. The final evaluation index calculation results are shown in Table 2.

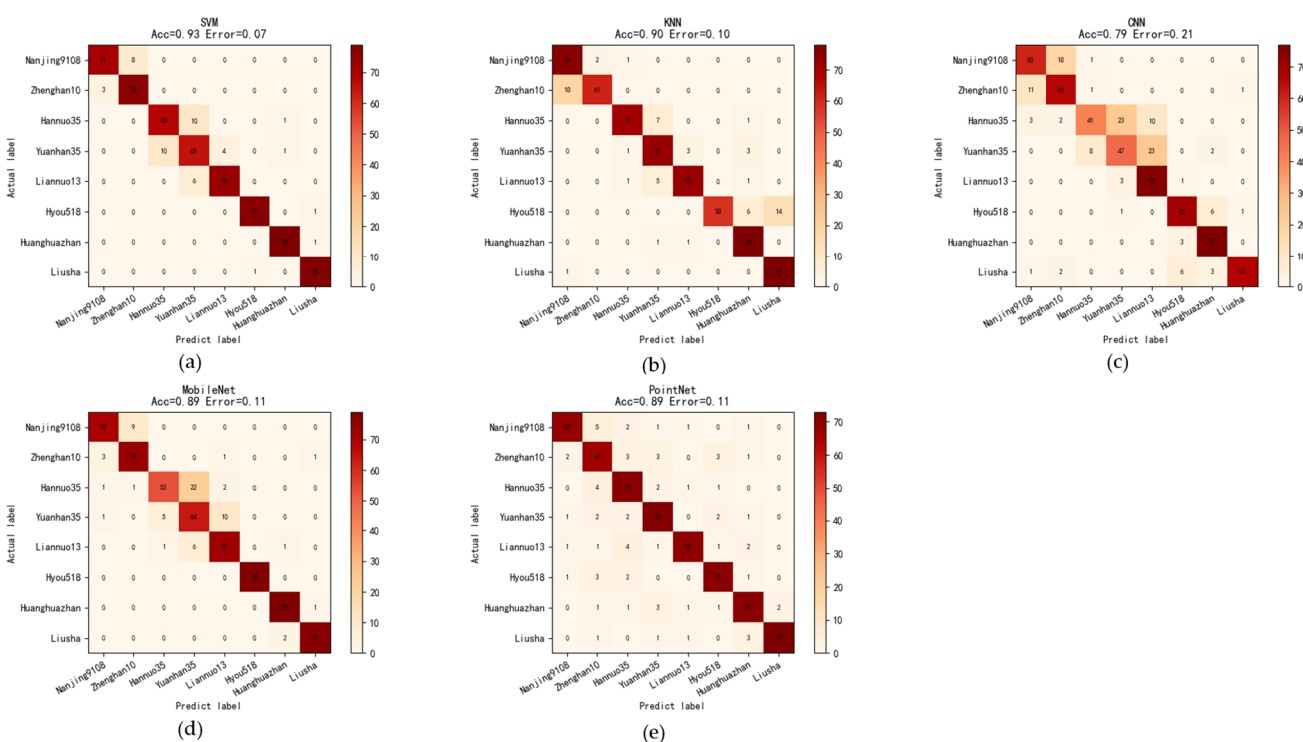

**Figure 12.** Confusion matrix. (**a**) SVM. (**b**) kNN. (**c**) CNN. (**d**) MobileNet. (**e**) PointNet.

**Table 2.** Measures to evaluate the model performance.

| Model | Macro-Precision | Macro-Recall | Macro-F1 |
|---|---|---|---|
| SVM | 0.928 | 0.929 | 0.929 |
| kNN | 0.908 | 0.894 | 0.894 |
| CNN | 0.801 | 0.795 | 0.790 |
| MobileNet | 0.899 | 0.894 | 0.894 |
| PointNet | 0.892 | 0.889 | 0.890 |

Macro-F1 was combined as the comprehensive scoring vector of the voting method. The SVM, kNN, CNN, MobileNet, and PointNet models predicted the probability of each sample in the test set as the input of the voting method. The models used the comprehensive scoring vector for weighted combination, and selected the recognition result with the highest probability as the final prediction value of late fusion. Table 3 shows the prediction results of all kinds of rice seed varieties after the final statistic of the improved voting method for late fusion. Using the improved voting method for late fusion, the final accuracy was 97.4%. Finally, the prediction accuracy for all varieties was more than 90%. Compared with the recognition accuracy of each model alone, the prediction results after fusion by the voting method were significantly improved. The recognition accuracy for Hyou518, Huanghuazhan, and Liusha was 100%. Although the accuracy for Hannuo35 was lower than that of the other varieties, it also reached 92.4%. The recognition accuracy of each variety before and after late fusion is compared in Table 3. It can be seen from the results that the improved voting method improved the accuracy of Hannuo35 and Yuanhan35 the most, from the average accuracy of the 2D recognition of 73.9% and 77.7% to the final accuracies of 92.4% and 96.3%, respectively. For these two kinds of rice, the classification effect of the 2D classification model is not ideal, but the classification accuracy of PointNet was relatively high. The 2D recognition effect was poor, for which there may be two reasons. One is that some 2D models (such as CNN and MobileNet) have poor classification effects on these two rice species. Second, the difference between the 2D images of these two rice species and other varieties is relatively small, so it will cause interference, and a large number of correct prediction results cannot be obtained. However, their differences in the 3D point cloud data were more obvious, so the 3D point cloud features can be used to classify them. For seeds such as Zhenghan10, Liannuo13, and Nanjing9108, PointNet's classification effect was not ideal, and the accuracy was lower than 89%. The reason may be that the differences in these seeds on the 3D point cloud were not very obvious and cannot provide a reliable basis for classification. However, they can be efficiently classified using the feature values of their 2D images.

**Table 3.** Accuracy comparison for identification of various rice varieties.

| Varieties | SVM | kNN | CNN | MobileNet | PointNet | Late Fusion |
|---|---|---|---|---|---|---|
| Nanjing9108 | 90.1% | 96.3% | 75.8% | 88.7% | 87.4% | 98.9% |
| Zhenghan10 | 96.1% | 77.4% | 84.1% | 93.8% | 85.0% | 97.4% |
| Hannuo35 | 86.2% | 90.3% | 52.1% | 67.0% | 89.0% | 92.4% |
| Yuanhan35 | 80.6% | 91.4% | 58.6% | 80.0% | 90.0% | 96.3% |
| Huanghuazhan | 93.2% | 91.3% | 94.7% | 90.3% | 87.5% | 97.5% |
| Hyou518 | 99.1% | 77.4% | 90.3% | 100.0% | 91.3% | 98.9% |
| Liannuo13 | 99.0% | 99.2% | 95.5% | 98.9% | 89.8% | 98.9% |
| Liusha | 98.5% | 99.4% | 84.8% | 97.4% | 90.0% | 98.9% |

Table 4 reflects the time taken by each model to predict the test separately and the time consumed to make predictions using late fusion. Comparing the late fusion time with the time consumption of the previous single model, it is not difficult to see that the time complexity caused by late fusion using the improved voting method mainly depends on the model it chooses. In other words, the more complex the model selected (such as

PointNet), the longer it takes. However, the voting method takes only 23.62 milliseconds, and it can be seen that it does not generate a large computational burden.

**Table 4.** Time-consuming comparison of various classification methods.

| Model | Prediction Time |
| --- | --- |
| SVM | 18.75 s |
| kNN | 19.98 s |
| CNN | 10.35 s |
| MobileNet | 20.76 s |
| PointNet | 9792 s |
| Late fusion | 9861.86 s |

By comparing this experiment with the existing research, it can be seen that the point cloud data obtained by using the light field camera can be used as an important basis for 3D classification. Furthermore, past studies have only focused on a single modality in 2D or 3D. However, the information contained in the two modalities can complement each other to classify from multiple perspectives. In the late fusion, this study replaces the process of calculating the average value in the traditional voting method by calculating the comprehensive score vector of each model separately. The prediction results of each model were weighted and fused by using the score vector. Experimental results showed that this fusion method can not only comprehensively evaluate multiple modalities, but also correct the prediction results of individual models with poor classification effects and error-prone models. Finally, the classification of rice seed varieties was effectively realized.

## 4. Conclusions

Based on the principle of multimodal fusion, we experimentally evaluated a rice variety classification method that used an improved voting method to perform late fusion of 2D and 3D modalities. The experimental data came from eight common rice varieties in China, and a Raytrix light field camera was used to collect 2D images and 3D point cloud data. We proposed an improved late-fusion method to generate a dynamically changing scoring vector according to the actual situation of the model, which was used to adjust the influence on the final prediction result. After preprocessing the data by noise reduction, filtering, and sampling, a dataset was obtained for classification. We input the dataset into models corresponding to the modality to obtain the predicted probabilities of the test set. The scoring vector was used to calculate the probability weighting of different models, and the predicted value with the highest final probability was selected as the final value. Compared with other multimodal fusion methods, this method was more robust. Its prediction results are not easily affected by a single model, and at the same time, it avoids possible interference from excessive model homogeneity or poor performance. The improved voting method was used to perform late fusion on the prediction results of the test set, and the final average accuracy reached 97.4%. Compared with a single SVM, kNN, CNN, MobileNet, and PointNet, the accuracy was 4.9%, 8.3%, 18.1%, 8.3%, and 9.0% higher, respectively. It can be seen that late fusion had the best effect on improving the accuracy of CNN and late fusion improved the identification accuracy of Hannuo35 and Yuanhan35 most obviously. The experimental results showed that the improved voting method can combine the advantages of different modal data and significantly improve the final prediction results.

This study provides a new perspective for the future classification of rice varieties. In future experiments, the rice seed dataset can be further expanded to provide sufficient data for the recognition algorithm. The preprocessing of point cloud data can be further optimized. The selection of points during sampling is not effective enough. Some important feature points may be deleted during preprocessing, which ultimately affects the classification results of the model. In addition, to pursue fast detection, the point cloud of

this experimental species is half a seed, and the next step is to register the point cloud data to obtain the complete seed data for training to obtain better classification results.

**Author Contributions:** Conceptualization, X.H., X.Z. and Y.Q.; Data curation, H.L.; Formal analysis, H.L.; Investigation, X.F.; Methodology, X.H., X.Z. and Y.Q.; Project administration, X.Z. and Y.Q.; Resources, W.Y.; Supervision, X.Z. and Y.Q.; Validation, X.H. and Q.C.; Visualization, Q.C.; Writing—original draft, X.H.; Writing—review & editing, X.H., X.Z. and Y.Q. All authors have read and agreed to the published version of the manuscript.

**Funding:** This work was funded by the National Natural Science Foundation of China Youth Fund Project (51305182); the Ministry of Agriculture Key Laboratory of Modern Agricultural Equipment (201602004); and the China University Industry-University-Research Innovation Fund Project (2021ZYA03010).

**Institutional Review Board Statement:** Not applicable.

**Data Availability Statement:** Not applicable.

**Conflicts of Interest:** The authors declare no conflict of interest.

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
