# Peer review of "Multi-Modal Late Fusion Rice Seed Variety Classification Based on an Improved Voting Method"

_agriculture, doi:10.3390/agriculture13030597_

Round 1

Reviewer 1 Report

The manuscript is written with clear understanding of the project addressed. However, there are major concerns that need to be addressed to enhance the quality of the manuscript. My specific comments are as follows:

Abstract: Add concluding remark.

Introduction:

Introduction is too long. Omit the part that are not really related to your study.

Based on your objectives, please compare how your study is different from those that have already been published.

Materials and Methods:

L156: ‘camera..” Add brand, state, country. Check for other equipment as well.

How about storage/treatment for rice seeds?

Spell out acronym for CNN at the first time you mentioned it

There is no information regarding the data analysis of the study. Please add.

Results and Discussion:

Since a comparison is being made between models, the requirements of each model and its computational needs, which lead to increased time requirements, should be discussed more extensively in the discussion.

Relate your results with existing literatures to support your findings. Instead of mentioning the results, the authors should justify/explain the findings

 Conclusion:

L463:” Compared with a single SVM, kNN, 463 CNN, MobileNet, and PointNet, the accuracy is 4.875%, 8.25%, 18.125%, 8.25%, and 9% higher, respectively.” Which produce the best result?

Add on main finding/results of the study. What are the main outcome based on the results. The authors should highlighted this matter.

General comments:

Please check the reference styles and grammar of the manuscript.

Author Response

Response to Reviewer 1 Comments

Point 1: Add concluding remark.

Response 1: We added a clear concluding remark in L21.

Point 2: Introduction is too long. Omit the part that are not really related to your study.

Response 2: We have omitted the overview of the whole field which is redundant in the first paragraph, and supplemented the introduction of relevant research on multimodal fusion.

Point 3: Based on your objectives, please compare how your study is different from those that have already been published.

Response 3: We added an explanation in the last paragraph of the introduction (L136), explaining the differences and innovations between our research and previous research. These include the use of light field cameras and the improvement of the late fusion voting method.

Point 4: L156: ‘camera..” Add brand, state, country. Check for other equipment as well.

Response 4: We added the countries, brands and models of cameras, lenses and other equipment in L158.

Point 5: How about storage/treatment for rice seeds?

Response 5: In the materials and methods section, we added a description of the scientific storage conditions of the rice seeds we used to show the reliability of storage and processing.

Point 6: Spell out acronym for CNN at the first time you mentioned it

Response 6: The first mention of CNN is in the introduction section (L58), and it is supplemented by its complete spelling.

Point 7: There is no information regarding the data analysis of the study. Please add.

Response 7: Because the input data of each model in this study were preprocessed image and point cloud data, there was no feature extraction. For better explanation, in L232 in Section 2.3, we supplemented the data analysis of the histogram and illustrated the information that can be obtained by using it as input data.

Point 8: Since a comparison is being made between models, the requirements of each model and its computational needs, which lead to increased time requirements, should be discussed more extensively in the discussion.

Response 8: In table 4 in the result and discussion section, we added the time-consuming of running each model separately. Then we compare the time consumption of the complete process of fusion using the voting method with them. In L467, we explained this phenomenon. Since the voting method needs to run each model first, its time consumption mainly depends on the time consumption and complexity of the selected model. However, excluding the time predicted by the model, the final fusion time only took 26 milliseconds, which shows that the time consumption of the voting method is not high.

Point 9: Relate your results with existing literatures to support your findings. Instead of mentioning the results, the authors should justify/explain the findings

Response 9: In L525, we added a comparison of our study with existing related studies and explained our findings in further detail in conjunction with experimental results.

Point 10: L463:” Compared with a single SVM, kNN, 463 CNN, MobileNet, and PointNet, the accuracy is 4.875%, 8.25%, 18.125%, 8.25%, and 9% higher, respectively.” Which produce the best result?

Response 10: In L511, we added that late fusion works best for CNN and analyzed this result.

Point 11: Add on main finding/results of the study. What are the main outcome based on the results. The authors should highlighted this matter.

Response 11: Starting at L511 of the conclusion section, we supplemented the explanation of the main outcome of the experiment, showing the advantages of late fusion from two perspectives of model and species.

Point 12: Please check the reference styles and grammar of the manuscript.

Response 12: We revised the syntax in the manuscript that was prone to ambiguity.

Reviewer 2 Report

Dear authors

This article aims to classify rice varieties using multi-modal late-fusion detection and improved voting methods. In general, it's an interesting article, and it is well written, but some points could be improved.

Here I presented some of my questions and suggestions to improve the article:

In the abstract:

-        Try to mention why it is crucial to classify the different rice varieties.

-        In (line 20), keep just one number after the point in the percentages.

-        Is it necessary to mention that there are 3194 varieties of rice?

-        Bring more conclusion to this section

In the Introduction section:

- Avoid writing very long sentences. Some of them have more than 30 lines. Split it into two or more.

- Describe the meaning of LR, MLP, DT, RF, and NB the first time they appear in the manuscript.

- What are RGB images? Describe it.

In the Materials and Methods section:

- In figure 1, change the place of the text inside the figure, put them on the white wall behind the computer, and change the direction of the arrows. It will improve the reading.

- Figure 4, you could increase this figure and improve its resolution.

- Line 310, avoid using "paper." It is not formal English.

- In figure 7, you could include the classifiers you used inside this flow chart.

- Who has used equations 1 to 5 for rice classification before? Please quote some articles.

In the Results and Discussion section

- Avoid writing very long sentences.

- Increase the quality of the figures.

- You present the results very well but should try to compare them with previous researchers so you can also improve your discussion.

In the conclusion section:

- (Line 464) keep just one number after the point in the percentages.

- Insert the information of lines 464-465 into the abstract section.

Best regards

Author Response

Response to Reviewer 2 Comments

Point 1: Try to mention why it is crucial to classify the different rice varieties.

Response 1: In the first part of the abstract, we supplemented the reasons why the classification of rice seeds is very important.

Point 2: In (line 20), keep just one number after the point in the percentages.

Response 2: We modified the original data to retain one decimal place.

Point 3: Is it necessary to mention that there are 3194 varieties of rice?

Response 3: There is ambiguity in the original expression here. What we actually want to express is that a total of 3194 seeds were collected as samples. We have removed this sentence and added the dataset size in the next sentence.

Point 4: Bring more conclusion to this section

Response 4: We added more conclusions in L21 (in revision mode) of the Abstract, including the accuracy of the model and the recognition effect of the voting method on different types of rice species.

Point 5: Avoid writing very long sentences. Some of them have more than 30 lines. Split it into two or more.

Response 5: For readability, we split longer sentences, such as L85, L112, L115, L126.

Point 6: Describe the meaning of LR, MLP, DT, RF, and NB the first time they appear in the manuscript.

Response 6: The LR, MLP, DT, RF, and NB mentioned in the manuscript are some algorithms, and we have added the full names.

Point 7: What are RGB images? Describe it.

Response 7: We added interpretation of RGB images on L92 in revision mode.

Point 8: In figure 1, change the place of the text inside the figure, put them on the white wall behind the computer, and change the direction of the arrows. It will improve the reading.

Response 8: We reshot the hardware environment in Figure 1. The shooting software has been added, and the position and direction of the arrow mark have been modified as required.

Point 9: Figure 4, you could increase this figure and improve its resolution.

Response 9: We chose a new way to save the histogram and replaced the original image in Figure 4, which has increased its resolution.

Point 10: Line 310, avoid using "paper." It is not formal English.

Response 10: We replaced the word 'paper' in L310 with 'study'.

Point 11: In figure 7, you could include the classifiers you used inside this flow chart..

Response 11: In Figure 7, we have supplemented the classifiers used in this experiment and classified it according to 2D and 3D.

Point 12: Who has used equations 1 to 5 for rice classification before? Please quote some articles.

Response 12: Equations 1 to 5 are a way to score models, but no one has applied them to rice species classification yet. In order to better illustrate the reliability of this method, we provided a supplementary explanation of this method and added references to this method to illustrate it.

Point 13: Avoid writing very long sentences.

Response 13: We have split the long sentences in the result and discussion section for easy reading.

Point 14: Increase the quality of the figures.

Response 14: In the result and discussion section we made tweaks to the data in an effort to improve the quality of the figures.

Point 15: You present the results very well but should try to compare them with previous researchers so you can also improve your discussion.

Response 15: In the last paragraph of result and discussion, we compared the existing research with our research, and analyzed and discussed it in combination with our experimental results. It reflects the innovativeness of the experiment we designed.

Point 16: (Line 464) keep just one number after the point in the percentages.

Response 16: In L464 we modified the data to keep just one number after the point in the percentages.

Point 17: Insert the information of lines 464-465 into the abstract section.

Response 17: We inserted the contents of the information of lines 464-465 into the abstract.

Round 2

Reviewer 1 Report

The authors have addressed all the comments. Hence, the paper can be accepted.